# 11β-HSD1 inhibition in men mitigates prednisolone-induced adverse effects in a proof-of-concept randomised double-blind placebo-controlled trial

Glucocorticoids prescribed to limit inflammation, have significant adverse effects. As 11β-hydroxysteroid dehydrogenase type 1 (11β-HSD1) regenerates active glucocorticoid, we investigated whether 11β-HSD1 inhibition with AZD4017 could mitigate adverse glucocorticoid effects without compromising their anti-inflammatory actions. We conducted a proof-of-concept, randomized, double-blind, placebo-controlled study at Research Unit, Churchill Hospital, Oxford, UK (NCT03111810). 32 healthy male volunteers were randomized to AZD4017 or placebo, alongside prednisolone treatment. Although the primary endpoint of the study (change in glucose disposal during a two-step hyperinsulinemic, normoglycemic clamp) wasn't met, hepatic insulin sensitivity worsened in the placebo-treated but not in the AZD4017-treated group. Protective effects of AZD4017 on markers of lipid metabolism and bone turnover were observed. Night-time blood pressure was higher in the placebo-treated but not in the AZD4017-treated group. Urinary (5aTHF+THF)/THE ratio was lower in the AZD4017-treated but remained the same in the placebo-treated group. Most anti-inflammatory actions of prednisolone persisted with AZD4017 co-treatment. Four adverse events were reported with AZD4017 and no serious adverse events. Here we show that co-administration of AZD4017 with prednisolone in men is a potential strategy to limit adverse glucocorticoid effects.

Glucocorticoids (GC) are prescribed to 2–3% of the population, most commonly for their anti-inflammatory actions[1]. Despite their therapeutic efficacy, GC use is limited by significant adverse effects including obesity, skeletal muscle myopathy, skin thinning, hypertension, osteoporosis, insulin resistance and type 2 diabetes (T2D), collectively termed 'Iatrogenic Cushing's syndrome'. Adverse effects are not limited to chronic use; frequent short-term GC administration is associated with increased morbidity and mortality[2]. The health burden associated with these adverse effects highlights the unmet clinical need to develop specific therapies that have the potential to ameliorate their detrimental side effect profile. Importantly, any potential therapeutic approach must not compromise the anti-inflammatory actions of GCs.

At a cellular level, the ability of GCs to bind to, and activate the GC receptor, is controlled by a series of enzymes including the 11β-hydroxysteroid dehydrogenases (11β-HSD1 and 2) that interconvert active and inactive GCs[3]. 11β-HSD2, is highly expressed in the kidney and rapidly inactivates cortisol to cortisone as well as synthetic prednisolone to inactive prednisone. Within metabolic tissues, including liver, skeletal muscle and adipose tissue, GCs are reactivated by

✉ e-mail: Jeremy.tomlinson@ocdem.ox.ac.uk

11β-HSD1, converting inactive cortisone to active cortisol and inactive prednisone to active prednisolone[4].

In preclinical studies, we have shown that 11β-HSD1 is critical in regulating the development of the adverse features associated with circulating GC excess. 11β-HSD1 knockout (KO) mice treated with high doses of GC fail to develop a classical Cushing's phenotype[5] and this is critically dependent upon adipose tissue 11β-HSD1 expression[5]. Endorsing these rodent data, we and others have reported patients with Cushing's disease and Cushing's syndrome, who were protected from the severe adverse effects of endogenous GC excess due to a functional deficit in 11β-HSD1 activity[6].

Highly potent and selective 11β-HSD1 inhibitors have been developed, initially as a potential therapy for patients with T2D and metabolic disease, although their beneficial effects on glucose levels were modest[7,8]. To date, their ability to limit the adverse effects of prescribed exogenous GC therapy has not been tested. AZD4017 is a potent, competitive inhibitor of human 11β-HSD1 that is safe and well-tolerated in clinical studies;[9] urinary steroid metabolite analysis has shown global inhibition of 11β-HSD1 to levels similar to those observed in patients with inactivating mutations in the HSD11B1 gene[10]. In addition, multiple other methodologies have been used to demonstrate the ability of AZD4017 to inhibit 11β-HSD1 activity including prednisolone generations curves, ex vivo biopsy assays as well as the use of isotopically labelled glucocorticoids[11–15]. Taking these data into account, only urine steroid metabolite assays were used to confirm enzyme inhibition in this study.

Based on the published literature that demonstrating the role of 11β-HSD1 to regenerate active GC as well as its tissue distribution (high levels of expression in metabolic tissues e.g. liver, adipose, skeletal muscle, and low levels of expression in immune-inflammatory cells), we hypothesised that selective and specific inhibition of 11β-HSD1 may represent an alternative strategy to limit the adverse effects of prescribed GCs without compromising their anti-inflammatory actions.

Adopting an experimental medicine, proof-of-concept approach, we have performed, a randomised, double-blind placebo-controlled study to test whether AZD4017 limits the adverse metabolic and bone effects of prednisolone (20 mg daily for 7 days) in healthy male volunteers.

## Results

42 male participants were assessed for study eligibility between 25th May 2017 and 13th February 2019; 10 failed the screening process leaving 32 participants who were enroled and randomised. Sixteen were assigned to AZD4017 (400 mg twice daily) plus prednisolone (20 mg daily) and 16 to placebo twice daily plus prednisolone (20 mg once-daily) (Fig. 1). One participant from the AZD4017 group was excluded due to <85% compliance with study medication and one participant from the placebo-treated group was excluded for repeated failure to fast before study procedures. Participant baseline demographics and biochemical characteristics are shown in Table 1.

### ACTH, serum and urine GC metabolites

At baseline, circulating cortisol and ACTH levels did not differ between the two groups (Table 1). Whilst post-treatment circulating cortisol levels were similar (91 [IQR 54–127] vs. 61 [IQR 36–150] nmol/L, $p = 0.55$, AZD4017 vs. placebo), post-treatment cortisone levels were significantly higher in the AZD4017-treated group (46.5 [IQR 35.8–55.9] vs. 20.0 [IQR 20.0–41.4] nmol/L, $p = 0.004$, AZD4017 vs. placebo). Post-treatment, ACTH levels decreased in the placebo (30.5 [IQR 16.9–54.6] vs. 11.7 [IQR 5.2–26.2] ng/L, $p = 0.030$) but not in the AZD4017-treated group (21.9 [IQR 18.1–52.9] vs. 20.4 [IQR 9.0–42.4] ng/L, $p = 0.22$). There was no significant difference in the post-treatment ACTH levels between the groups (20.4 [IQR 9.0–42.4] vs. 11.7 [IQR 5.2–26.2] ng/L $p = 0.23$, AZD4017 vs. placebo), nor any difference in the change in ACTH between the 2 groups ($p = 0.95$).

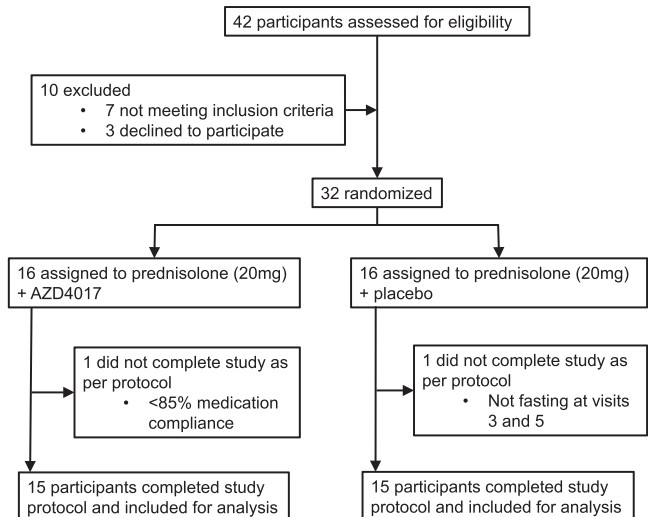

**Fig. 1 | Clinical trial flow chart.** Trial profile to determine the impact of selective 11β-hydroxysteroid dehydrogenase type 1 inhibition with AZD4017 to limit the adverse effects of prednisolone.

After 7 days of treatment, circulating prednisolone levels were similar in the AZD4017 and placebo-treated groups (650 ± 282 vs. 723 ± 430 nmol/L, p = 0.63, AZD4017 vs. placebo). Similarly, there were no differences in circulating prednisone levels (142 ± 62 vs. 103 ± 52 nmol/L, $p = 0.11$, AZD4017 vs. placebo) or the prednisolone/prednisone ratio (4.89 ± 2.48 vs. 6.33 ± 3.11, $p = 0.22$, AZD4017 vs. placebo).

Prior to treatment, the urinary (5aTHF+THF)/THE ratio (indicative of 11β-HSD1 activity) was similar in both groups (0.85 ± 0.19 vs. 1.04 ± 0.37, $p = 0.08$, AZD4017 vs. placebo). Consistent with 11β-HSD1 inhibition, after 7-days of treatment, the (5aTHF+THF)/THE ratio was significantly lower in the AZD4017 group (0.068 [IQR 0.059 to 0.086] vs. 0.99 [IQR 0.87 to 1.25], $p < 0.0001$, AZD4017 vs. placebo). The urinary cortisol/cortisone (reflecting 11β-HSD2 activity) ratio was similar in both groups at baseline and did not change following treatment (0.64 [IQR 0.57 to 0.76] vs. 0.71 [IQR 0.55 to 0.82], $p = 0.62$, AZD4017 vs. placebo).

### Glucose production and disposal

The primary endpoint of the study (change in stable isotope-measured Gd (glucose disposal) (low insulin) after 7 days of treatment), did not differ between the two groups (−0.54 [95% CI −1.87 to 0.57]; $p = 0.17$, AZD4017 vs. placebo). However, treatment with placebo+prednisolone significantly reduced Gd under low-dose insulin infusion (4.61 ± 2.28 vs. 3.06 ± 1.48 mg/kg min, $p = 0.009$, pre- vs. post-treatment), whereas it remained the same following co-administration of prednisolone with AZD4017 (4.59 ± 1.98 vs. 4.00 ± 2.65 mg/kg min, $p = 0.32$, pre- vs. post-treatment) (Supplementary Table 1). Similarly, glucose utilisation (corrected for circulating insulin levels, M/I-value) decreased following treatment with placebo+prednisolone during both low- and high-dose insulin infusions (Supplementary Table 1). In contrast, there was no change in M/I-value (low- or high-dose insulin) in AZD4017 + prednisolone treated group (Fig. 2, Supplementary Table 1): The absolute changes in Gd or M/I-value were not significantly different between the two arms of the study (Fig. 2, Supplementary Table 1).

Fasting glucose increased in the AZD4017 + prednisolone group, but not the placebo-treated group. In both groups, Ra Glucose and EGP increased after 7 days of treatment, with no significant between-group differences (Supplementary Table 1). Fasting insulin levels did not change in either group (Supplementary Table 1).

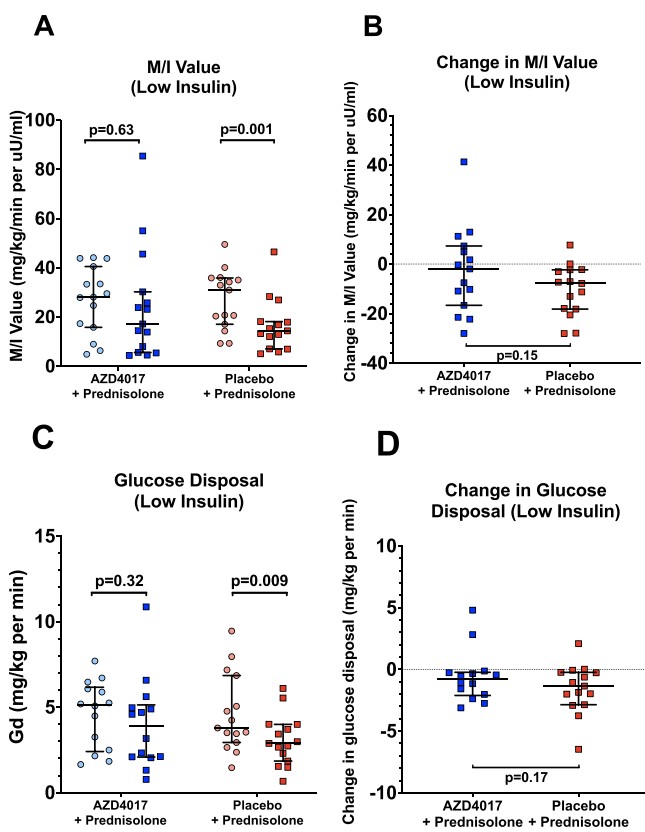

**Fig. 2 | Co-administration of the 11β-hydroxysteroid dehydrogenase type 1 inhibitor, AZD4017, limits the prednisolone-induced changes in insulin sensitivity (A and B) and glucose disposal (C and D) as measured across a 2-step hyperinsulinaemic euglycaemic clamp. A** Insulin sensitivity pre and post treatment in AZD4017+prednisolone and placebo+prednisolone groups. **B** Change in insulin sensitivity in AZD4017+prednisolone and placebo+prednisolone groups. **C** Glucose disposal pre and post treatment in AZD4017+prednisolone and placebo+prednisolone groups. **D** Change in glucose disposal in AZD4017+prednisolone and placebo+prednisolone groups. Data are medians and error bars are IQR. Data points represent individual patients. M/I value (Low insulin) AZD4017 + prednisolone n = 15, M/I value (Low insulin) placebo+prednisolone *n* = 15, Glucose disposal (Low insulin) AZD4017 + prednisolone n = 14, Glucose disposal (Low insulin) placebo+prednisolone *n* = 15. (Light blue circles = pre-treatment AZD4017 + prednisolone, blue squares = post-treatment AZD4017 + prednisolone, pink circles = pre-treatment placebo + prednisolone, red squares = post-treatment placebo + prednisolone). Statistical tests: Wilcoxon signed-rank test; Fig. 2A, generalised linear models adjusting for baseline variability in each specified outcome; Fig. 2B and D and paired two-tailed t-test; Fig. 2C.

## Circulating and adipose lipid metabolism

AZD4017 treatment prevented the prednisolone-induced increase in basal circulating triacylglycerol (TAG) levels. Furthermore, across all the phases of the 2-step hyperinsulinaemic euglycaemic clamp, the changes in circulating TAG levels associated with prednisone administration, were significantly reduced by AZD4017 (Supplementary Table 1, Fig. 3A, B and C).

Similarly, as expected, placebo+prednisolone caused an increase in circulating glycerol concentrations across all stages of the 2-step clamp. When prednisolone was co-administered with AZD4017, circulating glycerol concentrations did not change (Supplementary Table 1, Fig. 3D and E). Absolute changes in glycerol concentrations were not significantly different between the two groups (Supplementary Table 1, Fig. 3F).

Mirroring these observations, subcutaneous adipose interstitial fluid glycerol levels measured in adipose tissue microdialysis samples, increased in the placebo + prednisolone-treated group during low- and

**Table 1 | Baseline demographic, anthropometric and biochemical characteristics**

| Clinical Variable | AZD4017 (*n* = 15) | Placebo (*n* = 15) |
|---|---|---|
| Age, yr | 36.5 (11.0) | 39.0 (12.7) |
| Weight, kg | 79.9 (8.7) | 83.1 (8.5) |
| BMI, kg/m² | 24.5 (2.5) | 25.8 (2.0) |
| SBP, mm Hg | 135.7 (10.6) | 138.3 (14.4) |
| DBP, mm Hg | 78.5 (9.8) | 79.6 (11.4) |
| HbA$_{1c}$, mmol/mol | 34.0 (32.0–37.0) | 35.0 (33.0–36.0) |
| Fasting glucose, mmol/L | 4.6 (4.3–5.5) | 4.9 (4.4–5.1) |
| Fasting insulin, mU/L | 3.13 (1.79–6.60) | 2.84 (1.36–5.17) |
| HDL cholesterol, mmol/L | 1.3 (0.3) | 1.3 (0.3) |
| Total cholesterol, mmol/L | 5.0 (1.2) | 4.6 (0.9) |
| AST, IU/L | 21.3 (4.5) | 21.6 (6.7) |
| Bilirubin, mmol/L | 15.3 (5.3) | 15.0 (5.6) |
| ALT, IU/L | 21.0 (5.2) | 18.8 (5.5) |
| ALP, IU/L | 52.9 (8.9) | 57.3 (11.6) |
| Albumin, g/L | 39.5 (2.6) | 38.5 (2.3) |
| TSH, mIU/L | 1.4 (1.3–1.8) | 1.5 (1.0–1.9) |
| Androstenedione, nmol/L | 8.4 (5.9–9.4) | 6.4 (5.3–8.1) |
| DHEAS, μmol/L | 11.2 (5.5–15.1) | 6.7 (4.2–12.2) |
| SHBG, nmol/L | 33.4 (16.5) | 35.6 (15.9) |
| Testosterone, nmol/L | 18.4 (15.0–23.1) | 18.7 (13.6–22.0) |
| ACTH, ng/L | 21.9 (18.1–52.9) | 30.5 (16.9–54.6) |
| Cortisol, nmol/L | 372.7 (117.2) | 385.7 (77.1) |

Data are expressed as mean (SD) or median (IQR).
*ACTH* adrenocorticotropic hormone, *ALP* alkaline phosphatase, *ALT* alanine aminotransferase, *AST* aspartate aminotransferase, *BMI* body mass index, *DBP* diastolic blood pressure, *DHEAS* dehydroepiandrosterone sulphate, *DHT* dihydrotestosterone, *HbA1c* glycated haemoglobin, *HDL* high-density lipoprotein, *SBP* systolic blood pressure, *SHBG* sex-hormone binding globulin, *TSH* thyroid stimulating hormone.

high-insulin infusions. In the AZD4017 + prednisolone-treated group, there were no changes in adipose tissue interstitial glycerol levels pre- and post-treatment. There were no significant changes in the absolute changes in glycerol concentrations between the two groups (Supplementary Table 1, Fig. 3G, H and I).

Circulating NEFA levels were suppressed during low- and high-dose insulin infusions in both groups. High-dose insulin infusion failed to suppress NEFA levels in the placebo + prednisolone group (Supplementary Table 1). Ra palmitate decreased during high-dose insulin infusion following AZD4017 + prednisolone treatment, but there was no significant difference between the groups (Supplementary Table 1).

### Blood pressure
Although blood pressure measurements did not change significantly within each group, the absolute change in the mean night-time diastolic blood pressure was significantly higher in the placebo+prednisolone group (0.7 ± 8.1 vs. 4.6 ± 8.6, *p* = 0.03, AZD4017 vs. placebo) (Fig. 4). There were no changes in the mean day-time systolic and diastolic blood pressure, mean night-time systolic blood pressure (Supplementary Table 1).

### Bone turnover
Serum osteocalcin and P1NP are GC-sensitive markers of bone formation[16,17]. Osteocalcin levels decreased following administration of placebo + prednisolone (9.12 [IQR 7.44–9.76] vs. 4.16 [IQR 3.46–5.04] ng/ml, *p* < 0.0001, pre- vs. post-treatment) and this was entirely reversed with co-administration of AZD4017 (7.79 [IQR 6.18–11.69] vs. 8.13 [IQR 6.80–10.31]ng/ml, *p* = 0.64, pre- vs. post-treatment) (Supplementary Table 1, Fig. 5A). The absolute changes in osteocalcin levels were significantly different between the two groups

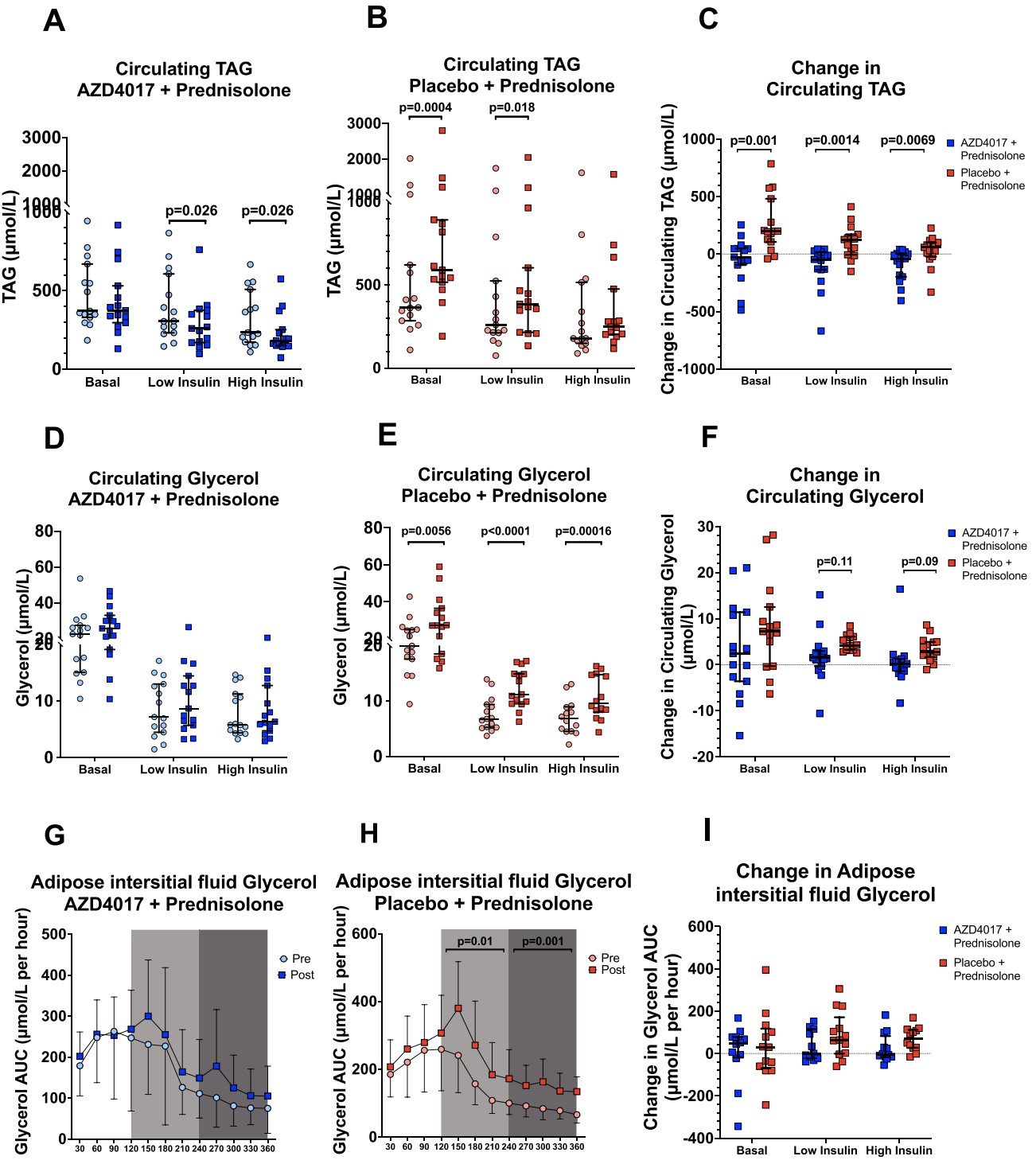

**Fig. 3 | Co-administration of the 11β-hydroxysteroid dehydrogenase type 1 inhibitor, AZD4017, limits the prednisolone-induced increase in circulating triacylglycerol (TAG) (A–C) and glycerol (D–F) levels measured across 2-step hyperinsulinaemic euglycaemic clamp. A** Circulating TAG levels pre and post treatment in the AZD4017+prednisolone group. **B** Circulating TAG levels pre and post treatment in the placebo+prednisolone group. **C** Change in circulating TAG levels in AZD4017+prednisolone and placebo+prednisolone groups. **D** Circulating glycerol levels pre and post treatment in the AZD4017+prednisolone group. **E** Circulating glycerol levels pre and post treatment in the placebo+prednisolone group. **F** Change in circulating glycerol levels in AZD4017+prednisolone and placebo+prednisolone groups. **G** Adipose interstitial fluid glycerol levels pre and post treatment in the AZD4017+prednisolone group. **H** Adipose interstitial fluid glycerol levels pre and post treatment in the placebo+prednisolone group. **I** Change in adipose interstitial fluid glycerol levels in AZD4017+prednisolone and placebo+prednisolone groups. Subcutaneous adipose tissue interstitial fluid glycerol levels

were increased by placebo + prednisolone treatment, but not when prednisolone was combined with AZD4017 (**G–I**). Data are medians, and error bars are IQR (**A–F, I**) and mean and SD (**G, H**). Data points represent individual patients (**A–F, I**) Circulating TAG AZD4017 + prednisolone: *n* = 15, Circulating TAG placebo + prednisolone: *n* = 15, Circulating Glycerol AZD4017 + prednisolone: *n* = 15, Circulating Glycerol placebo+prednisolone: Basal; *n* = 15, Low insulin; *n* = 15, High insulin; *n* = 14, Adipose interstitial fluid Glycerol AZD4017 + prednisolone: Basal; *n* = 12, Low insulin; *n* = 11, High insulin; *n* = 12, Adipose interstitial fluid Glycerol placebo + prednisolone: Basal; *n* = 13, Low insulin; *n* = 13, High insulin; *n* = 12, (Light blue circles = pre-treatment AZD4017 + prednisolone, blue squares = post-treatment AZD4017 + prednisolone, pink circles = pre-treatment placebo+prednisolone, red squares = post-treatment placebo + prednisolone). Statistical tests: Wilcoxon signed-rank test; Fig. 3A, generalised linear models adjusting for baseline variability in each specified outcome; Fig. 3 B, D and I and paired two-tailed t-test; Fig. 3C.

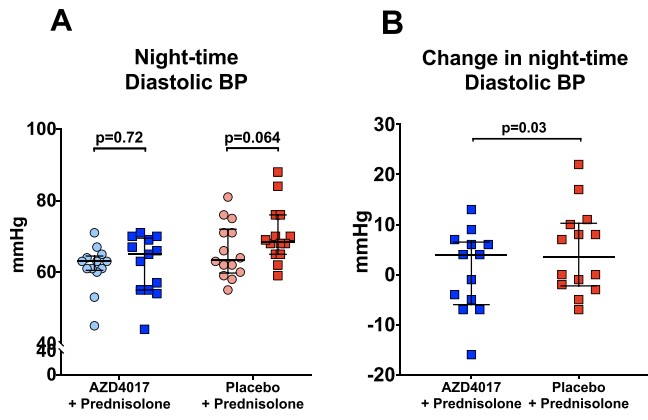

**Fig. 4 | The 11β-hydroxysteroid dehydrogenase type 1 inhibitor, AZD4017, prevents the rise in night-time diastolic blood pressure associated with 7 days of prednisolone + placebo treatment** (A and B). **A** Night-time diastolic blood pressure pre and post treatment in AZD4017+prednisolone and placebo+prednisolone groups. **B** Change in night-time diastolic blood pressure in AZD4017+prednisolone and placebo+prednisolone groups. Data are medians and error bars are IQR. Data points represent individual patients. AZD4017 + prednisolone; n = 13, placebo+prednisolone n = 14. (Light blue circles = pre-treatment AZD4017 + prednisolone, blue squares = post-treatment AZD4017 + prednisolone, pink circles = pre-treatment placebo+prednisolone, red squares = post-treatment placebo +prednisolone). Statistical tests: Wilcoxon signed-rank test; Fig. 4A, generalised linear models adjusting for baseline variability in each specified outcome; Fig. 4B.

(0.00 [IQR −3.46 to 2.18] vs. −3.86 [IQR −5.78 to −2.74], *p* < 0.0001, AZD4017 vs. placebo) (Supplementary Table 1, Fig. 5B). Mirroring our observations with osteocalcin, identical patterns in P1NP levels were observed, including significant differences in the absolute changes in P1NP between the two groups (Supplementary Table 1, Fig. 5C and D). CTX levels (as a marker of bone resorption) increased in the placebo +prednisolone group (0.44 [IQR 0.27–0.48] vs. 0.43 [IQR 0.35–0.59] ng/ml, *p* = 0.0028, pre- vs. post-treatment), but not in the AZD4017 + prednisolone group (0.51 [IQR 0.34–0.67] vs. 0.55 [IQR 0.37–0.71]ng/ml, *p* = 0.38, pre- vs. post-treatment) (Supplementary Table 1, Fig. 5E). The absolute changes in CTX levels were not significantly different between the two groups (0.04 [IQR −0.06–0.08] vs. 0.06 [IQR 0.003–0.14]ng/ml, *p* = 0.094, AZD4017 vs. placebo) (Supplementary Table 1, Fig. 5F).

## Immune function and cytokine profiling

The immune-suppressive action of prednisolone was measured using the OX40 assay. Treatment with both placebo+prednisolone and AZD4017 + prednisolone decreased the percentage of CD25 + CD134 + cells following exposal to both varicella zoster virus (VZV) and phytohemagglutinin (PHA) in the OX40 assay. Whilst there was no significant difference in response to VZV exposure when comparing the two arms of the study (−0.40 [IQR −0.55 to −0.10] vs. −0.30 [IQR −1.85–0.00], *p* = 0.81, AZD4017 vs. placebo), the response to PHA was more marked in the placebo+prednisolone group (−10.2 ± 14.4 vs. −21.7 ± 14.0, *p* = 0.046, AZD4017 vs. placebo) (Supplementary Table 1, Fig. 6A–D). In addition, targeted inflammatory serum proteomic analysis (O-link Target 48 Cytokine, O-Link, Uppsala, Sweden) revealed that 37/48 cytokines were quantifiable above the limits of assay detection. 19/37 were glucocorticoid-sensitive as evidenced by decreased levels following prednisolone +placebo administration (Supplementary Table 2). Of these 19 cytokines, only 5 (FMS-related receptor tyrosine kinase 3 ligand (FLT3LG), C-X-C motif chemokine ligand 10 (CXCL10), interferon-gamma (IFNG), Chemokine (C–C motif) ligand 19 (CCL19) and Matrix Metallopeptidase 12 (MMP12)) were differentially regulated by AZD4017 + prednisolone administration with significant differences

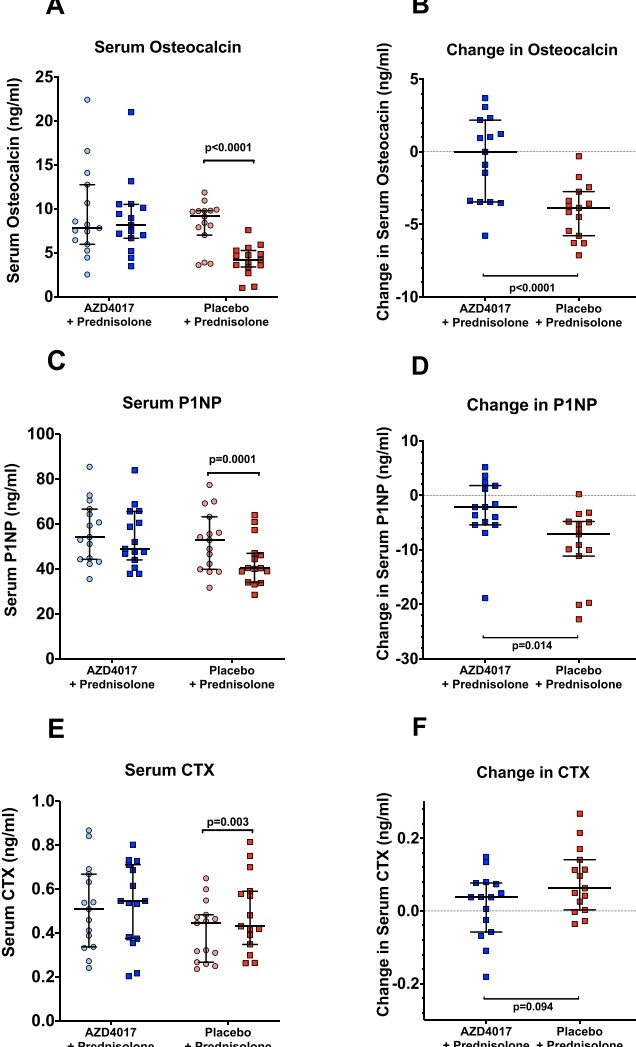

**Fig. 5 | The 11β-hydroxysteroid dehydrogenase type 1 inhibitor, AZD4017, prevents the decrease in serum osteocalcin** (A and B) **and P1NP** (C and D) **and increase in CTX** (E and F) **levels associated with 7 days of prednisolone+placebo treatment. A** Serum osteocalcin levels pre and post treatment in AZD4017+prednisolone and placebo+prednisolone groups. **B** Change in serum osteocalcin levels in AZD4017+prednisolone and placebo+prednisolone groups. **C** Serum P1NP levels pre and post treatment in AZD4017+prednisolone and placebo+prednisolone groups. **D** Change in serum P1NP levels in AZD4017+prednisolone and placebo+prednisolone groups. **E** Serum CTX levels pre and post treatment in AZD4017+prednisolone and placebo+prednisolone groups. **F** Change in serum CTX levels in AZD4017+prednisolone and placebo+prednisolone groups. Data are medians and error bars are IQR. Data points represent individual patients. AZD4017 + prednisolone; n = 15, placebo+prednisolone n = 15. (Light blue circles = pre-treatment AZD4017 + prednisolone, blue squares = post-treatment AZD4017 + prednisolone, pink circles = pre-treatment placebo+prednisolone, red squares = post-treatment placebo+prednisolone). Statistical tests: Wilcoxon signed-rank test; Fig. 5A, paired two-tailed t-test; Fig. 5C and E and generalised linear models adjusting for baseline variability in each specified outcome; Fig. 5B, D and F.

in response to treatment comparing each arm of the study (Fig. 6E, Supplementary Table 2).

## Adverse effects

There were four adverse events reported in the AZD4017 group, comprising transient headache (three participants) and an elevated TSH (one participant) which normalised by day 14. There was one reported adverse event in the placebo group (irritability). No other adverse events were reported throughout the duration of the study, or

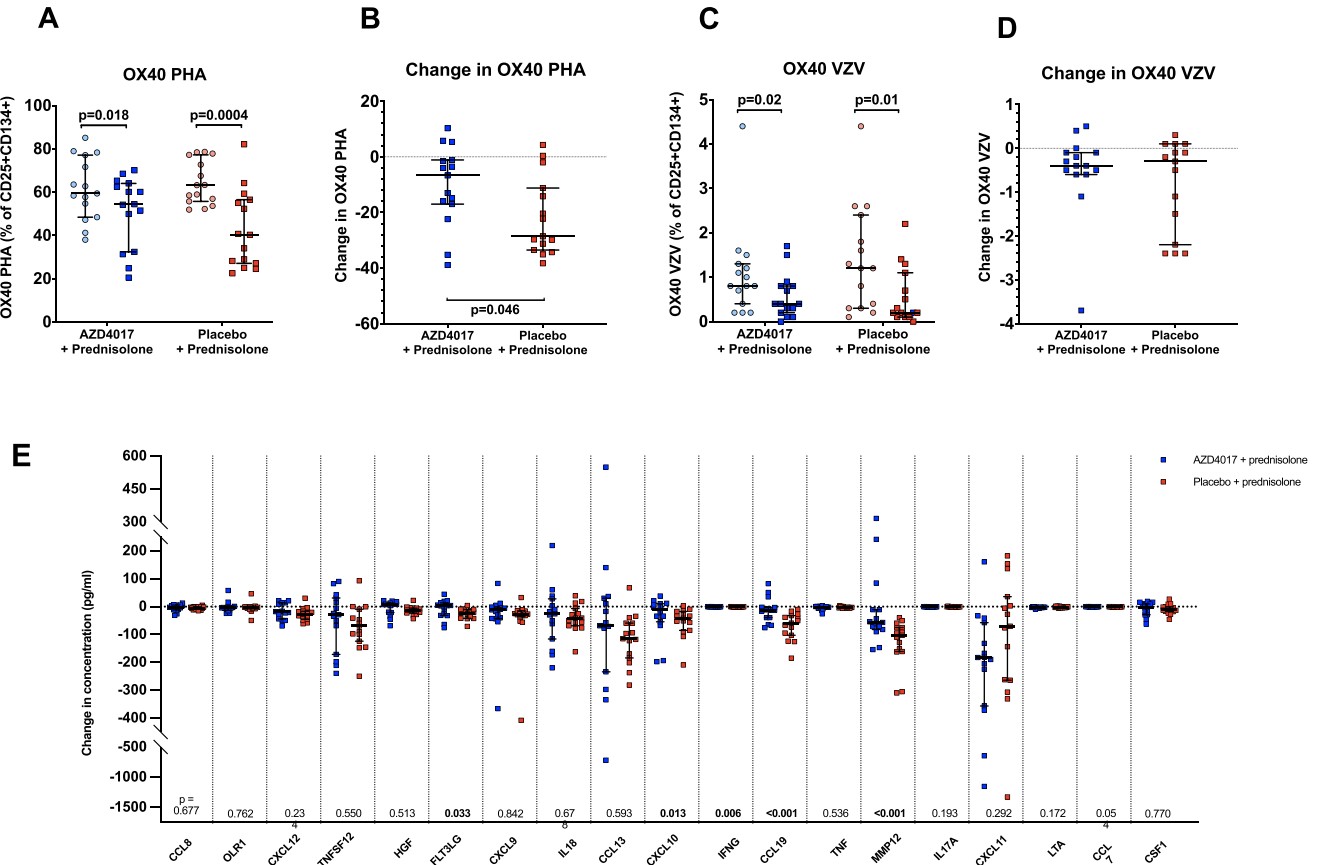

**Fig. 6 | The 11β-hydroxysteroid dehydrogenase type 1 inhibitor, AZD4017, does not prevent the reduction in OX40 assay readout (% of CD25 and CD134 positive cells following phytohemagglutinin (PHA) or varicella zoster (VZV) stimulation) associated with 7 days of prednisolone treatment** (A–D). **A** OX40 PHA levels pre and post treatment in AZD4017+prednisolone and placebo+prednisolone groups. **B** Change in OX40 PHA levels in AZD4017+prednisolone and placebo+prednisolone groups. **C** OX40 VZV levels pre and post treatment in AZD4017+prednisolone and placebo+prednisolone groups. **D** Change in OX40 VZV levels in AZD4017+prednisolone and placebo+prednisolone groups. Changes

in glucocorticoid responsive circulating cytokines following treatment with placebo+prednisolone vs. AZD4017 + prednisolone (**E**). Data are medians and error bars are IQR. Data points represent individual patients. AZD4017 + prednisolone; *n* = 15, placebo+prednisolone *n* = 15. (Light blue circles = pre-treatment AZD4017 + prednisolone, blue squares = post-treatment AZD4017 + prednisolone, pink circles = pre-treatment placebo+prednisolone, red squares = post-treatment placebo+prednisolone). Statistical tests: Wilcoxon signed-rank test; Fig. 6A and c, and generalised linear models adjusting for baseline variability in each specified outcome; Fig. 6B and D.

in the 30 days after the final administration of AZD4017 or placebo. There were no serious adverse effects or discontinuations due to adverse effects in either group.

## Discussion

This is a randomised, double-blind, placebo-controlled study, to show that inhibition of 11β-HSD1 using AZD4017 in male participants has the potential to limit the adverse metabolic and bone effects of oral prednisolone treatment. Although this is a small, proof-of-concept study, it highlights the role of 11β-HSD1 to regulate the action of exogenous GCs in clinical practice. In addition, it suggests that there is a critical role of tissue-specific GC metabolism (inactivation and regeneration), over and above circulating levels in the blood, in controlling the biological impact of GCs. Prednisolone metabolism (inactivation and reactivation) occurs intracellularly, within key target organs and therefore circulating levels do not provide an accurate reflection of tissue- and cell-specific exposure to active/inactive glucocorticoids. We did not observe alterations in circulating prednisolone or prednisone levels, but these measurements are not able to identify the precise balance of prednisone/prednisolone to which a specific cell or tissue is exposed, and our data would suggest that it is this which is critical to the development of adverse effects.

Previous studies with selective 11β-HSD1 inhibitors, including AZD4017, have shown activation of the HPA axis with increased ACTH

as a consequence of decreased peripheral cortisol generation[13,15]. The interpretation of ACTH values from the current study is challenging (only a single time point was measured) as prednisolone will suppress ACTH. Whilst ACTH decreased in the placebo-treated group, ACTH was unchanged in the AZD4017 group suggesting a potential lack of HPA axis suppression (but no activation). There were no differences between groups and additional studies would be needed to test whether there is the potential for 11β-HSD1 inhibitors to reduce the risk of prednisolone-induced adrenal suppression.

The metabolic effects of 11β-HSD1 deficiency or inhibition have been extensively investigated in animal models. 11β-HSD1 KO mice have reduced visceral adipose tissue accumulation on a high fat diet and resist the development of insulin resistance and T2D[18]. Selective 11β-HSD1 inhibitors have also been investigated in animal models demonstrating they reduce fasting blood glucose, insulin and cholesterol levels[19,20], as well as decreasing mesenteric fat mass and hepatic steatosis[21]. In addition, they improve insulin sensitivity and reduce hepatic glucose production[22]. However, in humans, selective 11β-HSD1 inhibitors, albeit given for only 12 weeks, have only demonstrated relatively modest improvements in glycaemic control, hyperlipidaemia and blood pressure in patients with T2D when taken for 12 weeks[3,7,8,23].

Cushing's syndrome as a result of excess exogenous GC therapy is a significant clinical and economic burden. The use of prescribed

glucocorticoids is widespread, particularly amongst the elderly who may be more susceptible to their adverse effects, and is associated with increased cardiovascular and cerebrovascular morbidity and mortality[24].

To date, the potential to modulate the adverse effects of prescribed GCs through manipulation of 11β-HSD1 activity and expression has only been examined in rodent models. 11β-HSD1 KO mice fail to develop a classical Cushing's phenotype following corticosterone (the predominant circulating GC in rodents) administration; this was critically dependent upon adipose tissue (but not liver) 11β-HSD1 expression[5]. More recently, Fenton et al. reported protection from GC-induced trabecular bone loss, in 11β-HSD1 KO mice[25]. The phenotype observed in these rodent models mirrors the observations from our clinical study in terms of the beneficial impact of 11β-HSD1 inhibition to limit the impact of prednisolone on glucose disposal, insulin sensitivity, blood pressure and osteocalcin.

The expression and activity of 11β-HSD1 in human bone along with its potential role in bone biology are extensively described in the literature[4,26–29]. Short-term administration of oral GCs has a potent impact on bone turnover markers in healthy male individuals, decreasing osteocalcin and P1NP and increasing CTX[16]. Our data represent an interventional study of 11β-HSD1 inhibition in humans, receiving systemic GC therapy. However, Cooper et al. examined the effect on bone formation and turnover markers following the administration of prednisolone (5 mg twice daily). Whilst, bone formation markers decreased in all subjects, the magnitude of the reduction was greatest in those individuals with the highest 11β-HSD1 activity at baseline[30]. The data from our study would appear to support these observations. Prednisolone-induced decreases in bone formation (osteocalcin and P1NP) and increases in bone resorption (CTX) were prevented by co-administration of AZD4017. As expected, the relative decrease in osteocalcin with prednisolone (−123%) was greater than P1NP (−22%), reflecting the additional direct impact of GCs on the osteocalcin gene transcription in addition to their effects on bone formation[31]. A recent study demonstrated no impact of 11β-HSD1 inhibition with AZD4017 on bone loss in post-menopausal women[32] and, therefore, the benefits that we observed would seem to relate specifically to glucocorticoid-induced bone loss.

Overall, the data from this study would support the concept that 11β-HSD1 inhibition may have potential as a therapy to limit GC-induced bone loss. However, larger studies incorporating more detailed bone phenotyping to confirm and endorse these data are now warranted.

GCs are most commonly prescribed for their immune-suppressive and anti-inflammatory actions. 11β-HSD1 is highly expressed in the liver and adipose tissue, but is only expressed at very low levels in T-cells and other immune-inflammatory response cells. It has been suggested that 11β-HSD1 may have a role in controlling the endogenous anti-inflammatory response; 11β-HSD1 KO mice develop a greater inflammatory reaction in response to joint inflammation, peritonitis and lung inflammation and have a slower recovery[33]. However, the response to exogenous GC therapy in these models in 11β-HSD1 KO mice has not been examined.

The OX40 assay is used in clinical laboratories to assess immune (T-cell) function and relies upon the flow cytometric detection of the upregulation of CD25 and OX40 (CD134) on the cell surface following incubation with antigen. The OX40 assay was selected on a proof-of-principle basis as there is evidence of glucocorticoid-induced suppression of activity (endorsed by Fig. 6A and C). Our data demonstrate that the impact of prednisolone OX40 response is largely preserved following co-administration of AZD4017. The serum proteomic analysis would also suggest that major components of the immune inflammatory responses to prednisolone persist with AZD4017 co-administration. The glucocorticoid-sensitive responses to only 5/19 cytokines were compromised by AZD4017, but the clinical significance

of this observation remains to be determined. We would speculate that the differential impact of AZD4017 metabolic and immune-inflammatory cells is a reflection of variable levels of expression. In tissues where 11β-HSD1 is highly expressed, circulating levels are amplified through the conversion of prednisone to prednisolone, whereas in tissues and cells where there is little 11β-HSD1, the impact of GCs reflect circulating levels (and hence no difference between placebo-treated and AZD4017-treated groups in our study). It is important to note that the participants in this study were healthy volunteers without any underlying immune or inflammatory conditions. Whilst these data do not allow us to be categorical about the preservation of the anti-inflammatory response, they do provide some reassurance that the majority of prednisolone-induced cytokine changes are preserved. However, this does not negate the need to undertake dedicated studies in patients with inflammatory conditions.

A small number of clinical studies have tried to address the issue of protection from the adverse metabolic effects of prescribed GCs, most commonly using established anti-diabetic agents. Glucagon-like peptide 1 (GLP-1) analogues appear to limit dysglycaemia associated with prednisolone use as well as improving pancreatic islet cell function[34]. In case series and retrospective analyses, dipeptidyl peptidase-4 (DPP-4) inhibitors improved glycated haemoglobin and post-prandial glucose levels in patients with chronic medical conditions treated with GC therapy[35]. However, in a randomised, double-blind placebo-controlled study, there was no improvement in GC-induced glucose intolerance in non-diabetic individuals[36].

More recently, Pernicova et al. examined the impact of metformin in patients already prescribed glucocorticoid therapy[37]. As well as reporting improvement in carbohydrate and lipid metabolism, they demonstrated improvements in the incidence of pneumonia and all-cause hospital admissions. Metformin administration improved some of the metabolic profiles of GC-treated patients. However, the participants in this study were all already taking GC therapy (and there was little deterioration in metabolic phenotype in the placebo group over time) and therefore it is not possible to say if the observed benefits of metformin were independent of the effects of GC therapy.

A key strength of the study is that the impact of a selective 11β-HSD1 inhibitor to treat iatrogenic Cushing's syndrome has not been explored before. Additional strengths include the study design and the use of state-of-the-art metabolic phenotyping. There are however some significant limitations; the sample size is modest, although the trial was designed as an early phase II, proof-of-concept study. As a consequence of the small sample size, subtle, non-significant differences in baseline characteristics could potentially impact outcome. To correct for this, the statistical models used in the analysis adjust for any differences in baseline variables between groups. In addition, the study was only conducted in healthy male volunteers and over a relatively short duration. We can therefore not draw conclusions with regards to the potential benefits in women (although the drug has subsequently been administered to women for up to 12 weeks, and was well tolerated)[9] and there is a need to undertake similar studies in women. Furthermore, the age of the population studied was relatively young (mean age 38 years) and therefore it cannot be assumed that these data would extrapolate to a more elderly population and additional studies, including in patients taking prescribed GCs for inflammatory conditions, are now needed.

The variability in Gd at baseline was higher than we had anticipated and higher than the values that we have used to inform our power and sample size estimates. This higher-than-anticipated variability in Gd may possibly explain why we were unable to detect significant differences in Gd between the two arms of the study despite having performed appropriate sample size and power calculations. Therefore, we cannot draw any robust conclusions as to the ability of AZD4017 to mitigate the adverse effects of prednisolone on glucose metabolism (although the patterns observed with regards to M/I-value,

Gd as well as endogenous glucose production rates are all consistent). Further studies, in larger populations, perhaps using alternative measures of glucose metabolism, with adequate power are undoubtedly warranted. Importantly, the participants in this study did not have underlying immune or inflammatory conditions and therefore, the next step would be to conduct a larger, more prolonged trial in patients with underlying inflammatory conditions who are initiated on glucocorticoid therapy and who are more likely to suffer the long-term adverse side effects.

11β-HSD1 inhibition appears to convey multiple metabolic benefits; we propose that this results from the specific targeting of a critical component of the tissue-specific action of GCs, whereby their impact in metabolic tissues is amplified (as a result of high 11β-HSD1 expression and active GC regeneration) leading to unwanted adverse side effects. We have demonstrated, that 11β-HSD1 inhibition, using AZD4017, has the potential to limit the adverse metabolic and bone side effects of prednisolone in healthy male participants. In the future, it may be that 11β-HSD1 inhibitors could prove to be an important protective adjunctive therapy alongside prescribed GCs and in addition may also offer another treatment strategy for patients with endogenous GC excess.

## Methods

### Study design and setting
We conducted a randomised, proof-of-concept, double-blind, placebo-controlled clinical study, undertaken at the Clinical Research Unit (CRU), Churchill Hospital (Oxford, UK). This clinical trial was registered at ClinicalTrials.gov (NCT03111810).

### Aims
The primary endpoint was the change observed in glucose disposal (Gd) from pre-treatment measurement to post-treatment assessment, as measured during a hyperinsulinaemic clamp. Pre-defined secondary endpoints included changes in EGP, lipolysis (systemic and adipose tissue), 24 h BP measurements, circulating lipid profiles, osteocalcin, urinary steroid metabolites, gene expression profiles and immunoinflammatory response as measured by the OX40 assay.

### Participants
Healthy male volunteers were recruited from local advertisement and from the Oxford Biobank (reference 08/H0606/107). Participants were recruited if they met the following inclusion criteria: male, aged 18–60 years with a BMI of 20-30 kg/m$^2$. Key exclusion criteria included, hypertension, hypercholesterolaemia, use of GC therapy within the last 6 months, taking medications known to impact upon GC metabolism, eGFR <60 mL/min/1.73 m$^2$, abnormal liver chemistry and diabetes mellitus. Diabetes mellitus was excluded on the basis of an HbA1c < 48 mmol/mol and fasting glucose <7.0 mmol/L (126 mg/dl) in accordance with internationally established criteria. Oral glucose tolerance tests were not performed. Written informed consent was obtained from all participants.

In this early phase study, the decision was made to recruit only male participants to avoid the complicating issues of changes in metabolic phenotype across the menstrual cycle or through the use of hormonal contraception. This is important, not only due to the impact of oestrogen on metabolic phenotype, but also its impact on steroid hormone binding globulins that have the potential to modify GC action. In addition, due to the stringent requirements for the necessity of contraception in female participants that may have included hormonal treatment that could not be stopped and may impact upon metabolic outcomes, only male participants were recruited.

### Randomisation and masking
Participants were randomised 1:1 to treatment with AZD4017 (400 mg PO twice-daily) or matching placebo, in addition to oral prednisolone (20 mg once-daily), according to an independently developed randomisation table, using blocks of four. Participants were randomised sequentially according to their recruitment date. Both participants and investigators were masked to the treatment allocation.

### Procedures
Participants attended the CRU at 08.00 h after an overnight fast (from 24.00 h). Fasting blood samples were taken between 08.00 h and 09.00 h for biochemistry, endocrine, lipid profiles, inflammatory cell and markers of bone turnover. Participants were fitted with a 24-h ambulatory blood pressure monitor (Microlife WatchBP 03 ABPM). Finally, they were provided with a container to perform a timed overnight urine collection (22.00 h–08.00 h) for assessment of urine steroid metabolites.

Participants returned to the CRU the following day having fasted from 24.00 h the previous night. At 08.00 h, a microdialysis catheter (CMA Microdialysis, Solna, Sweden) was inserted under local anaesthetic (2 mL, 1% lidocaine) into the subcutaneous adipose tissue, 5–10 cm lateral to the umbilicus in order to allow sampling of adipose tissue interstitial fluid. Using a microdialysis pump (CMA 106), the microdialysate solution (physiological sterile saline solution) was introduced into the catheter (perfusion rate = 0.3 μl/min) and samples were collected at 30-min intervals until the completion of the 2-step hyperinsulinaemic euglycaemic clamp (see below).

### 2-step hyperinsulinaemic euglycaemic clamp
At the commencement of the two-step hyperinsulinaemic euglycaemic clamp, basal unenriched blood samples were taken. A bolus of [U-$^{13}$C]-glucose (Cambridge Isotope Laboratories, Andover, USA) was administered (2 mg/kg over 1 min followed by a continuous infusion in 0.9% saline (20 μg/kg/min) as well as a continuous [2,2-$^2$H$_2$]-palmitate infusion (Cambridge Isotope Laboratories, Andover, USA) (0.03 μmol/kg/min) made up in human serum albumin. Blood glucose was monitored at 15-min intervals during the initial 120 mins ($t = 0$–120 min, basal phase). At $t = 120$ min, an insulin infusion (Actrapid; Novo Nordisk) was infused at 20 mU/m$^2$/min (low-dose) alongside an infusion of 20% dextrose supplemented with [U-$^{13}$C]-glucose to 4%; blood glucose levels were monitored at 5-min intervals ($t = 120$–240 min). At $t = 240$ min, the insulin infusion rate was increased to 100 mU/m$^2$/min (high-dose) and continued for another 120 min ($t = 240$–360 min) with continued 5-min blood glucose monitoring. Blood samples were taken at 3 time points in the last 30 min of each phase (basal, low- and high-insulin) for steady-state measurements of insulin, whole body glucose turnover (Ra glucose, Gd glucose), endogenous glucose production rate (EGP) and lipolysis (Ra palmitate). Glucose and palmitate disposal rates were calculated using a modified version of the Steele equations[38,39].

Participants were then randomised to receive either AZD4017 400 mg BD (AstraZeneca, UK) and prednisolone 20 mg daily or matched placebo and prednisolone 20 mg daily, for 7 days. They returned on the 6th day of administration for repeat fasting blood tests, 24 h blood pressure and urine collection and also on the 7th day of administration for a repeat 2-step hyperinsulinemic euglycemic clamp and microdialysis. On each day of investigation, trial medications were taken before the procedures were performed (Supplementary Fig. 1). Details of currently available pharmacokinetic and pharmacodynamic data relating to AZD4017 as well as dose justification (for AZD4017 and prednisolone) are presented in the supplementary data.

### Biochemical and stable isotope analysis
Cholesterol, liver biochemistry and plasma glucose were measured using standard laboratory methods (Roche Modular system, Roche Ltd, Lewes, UK). Insulin and osteocalcin were measured using a commercially available colorimetric ELISAs (Insulin; Mercodia, Uppsala, Sweden, Osteocalcin; Thermofisher, Frederick, USA). Additional bone

markers (type I N-terminal propeptide of type I procollagen (P1NP) and C-terminal telopeptide of type I collagen (CTX)) were analysed using the Roche Cobas e411 auto-analyser (Roche Diagnostics, Penzberg, Germany). The interassay CVs for P1NP and CTX were 5.1% and 4.9%, respectively. Serum samples were stored at −80°C and analysed together in the same assay run after one freeze-thaw cycle. Micro-dialysis samples were analysed for glycerol levels, using a mobile photometric, enzyme-kinetic analyser (CMA ISCUS Flex, Solna, Sweden).

Plasma enrichment of [U-$^{13}$C]-glucose was measured using gas chromatography-mass spectrometry. Plasma lipids were extracted using the Folch method and the non-esterified fatty acid (NEFA) fraction isolated by solid phase extraction (Bond Elut NH2- Aminopropyl columns). Following methylation of the NEFA fraction, the samples were run on GC (model 5890; Agilent Technologies, Cheshire, UK) to determine the relative amount of individual fatty acids. Whole body lipolysis (Ra palmitate) was calculated using extracted total circulating lipid and by measuring deuterium ([2,2-$^2$H$_2$]-palmitate) enrichment in the NEFA fraction. [2,2-$^2$H$_2$]-palmitate enrichments were determined by gas chromatography-mass spectrometry using a 5890 GC coupled to a 5973N MSD (Agilent Technologies; CA, USA). Ions with mass-to-charge ratios ($m/z$) of 270 (M + 0) and 272 (M + 2) were determined by selected ion monitoring.

## OX40 assay
Heparin blood was collected from participants pre-treatment and 7 days post-treatment and diluted 1:1 with Iscoves modified medium (Sigma-Aldrich). Each sample was tested for CD4 + T cell-specific responses towards Phytohaemagluttinin (PHA) (Sigma-Aldrich) and Varicella-Zoster virus (VZV) (Source-Bioscience). 2 µg/ml of antigen or 15 µg/ml of PHA was added to a 500 µl volume with appropriate negative control tubes. After incubation at 37 °C, 5% CO$_2$ for 42–44 h, samples were vortexed and 100 µl was aliquotted to a new tube. Blood was then stained with the following monoclonal fluorochromes; CD25–FITC, CD134-PE, CD14-PerCP, Viaprobe (7-AAD), CD4-APC and CD3-PE-Cy7 (BD Bioscience). Samples were incubated for 15 min at room temperature before having 1 ml of FACSLyse (BD Bioscience) solution added for 5 min. Samples were washed twice with PBS before being fixed with 1% paraformaldehyde and run on a BD FACSCanto II (BD Bioscience). Analysis was carried out using BD FACSDiva software (version 6.1; BD Biosciences).

Samples were gated to acquire 30,000 CD3 positive events, which were then negatively gated for CD14 and 7AAD. Quantification of CD3 + CD4 + CD25 + CD134 + events were calculated by setting gate coordinates on the media-only sample to equal 0.1% CD3 + CD4 + CD25 + CD134 + events as a percentage of all CD3 + CD4 + events. These coordinates were applied to subsequent antigen tubes to produce a percentage value of CD25 + CD134 + double-positive events[40]. Values pre- and post-treatment were compared to determine a delta-change percentage of antigen-specific CD4 + T cells.

## Serum cytokine proteomic analysis
Fasting plasma samples pre- and post-treatment were analysed using the O-link proximity extension assay platform (Olink® Target 48 Cytokine panel, O-link, Uppsala, Sweden) which has been validated against other cytokine multiplex platforms (https://www.olink.com/application/multiplex-analysis-of-inflammatory-proteins-2/). Only samples with cytokine levels above the limits of detection of the assay were included in the analysis.

## Serum and urine steroid metabolites
Prednisolone, prednisone, cortisol and cortisone were extracted from participants' serum, calibration standards and quality control by liquid:liquid extraction using diethyl ether. The upper solvent layer containing the steroids was separated from the bottom aqueous layer

using an ice-bath (Thermofisher) filled with ethanol. A rotary evaporator was used to dry down extracts and they were then reconstituted in 50:50 methanol-water before analysis on the mass spectrometer.

Total urinary cortisol, cortisone, THF, alloTHF and THE were measured by using a validated high-performance liquid chromatography tandem-mass spectrometry assay[41]. For all components, appropriate internal standards were added and the mixtures were incubated with an enzyme solution consisting of sulfatases and β-glucuronidases, to ensure hydrolysis of cortisol and the metabolites from their sulfated and glucuronidated forms. Internal standards that were used were cortisol-13C3, cortisone-D7, THF-D5, THF-D5 and alloTHF-D5. Subsequently, the analytes were extracted using a Supported Liquid Extraction technique. Finally, separation and detection were performed by use of a Phenomenex Luna Phenyl-Hexyl column (particle size 3 µm, 2.0 mm internal diameter by 150 mm; Waters) and a XEVO TQ-s tandem mass spectrometer operated in negative electro-spray ionisation mode (Waters). Intra- and inter-assay variation coefficients were <5.7% and <9.8%, respectively.

## Outcomes of interest
The primary endpoint was the change observed in glucose disposal (Gd) from pre-treatment measurement to post-treatment assessment, as measured during a hyperinsulinaemic clamp (low-dose insulin). The decision used Gd at low insulin infusion rates was made to try and balance the counteracting effects of prednisolone against the very potent impact of insulin. Published data have shown that prednisolone suppresses Gd at low dose, but not high-dose insulin[42]. Pre-defined secondary endpoints included changes in EGP, lipolysis (systemic and adipose tissue), 24 h BP measurements, circulating lipid profiles, osteocalcin, urinary steroid metabolites and immune-inflammatory response as measured by the OX40 assay. Changes in whole-body oxidation, in body composition and in adipose tissue and skeletal muscle gene expression profile were not included in this manuscript. These data will be incorporated into, and published in a subsequent manuscript.

## Statistical analysis
Reflecting the nature of this proof-of-concept, experimental medicine study, robust clinical date upon which to formulate accurate sample size estimates are lacking. Data from euglycaemic clamps suggest that pre-dnisolone may cause a >20% decrease in glucose disposal[34]. The study was therefore powered to detect a 20% difference in glucose disposal between the 2 arms from a typical mean (±1 SD) baseline of 2.91 ± 0.52 mg/kg/min with a power of 0.8 and a type 1 error of 0.05;[42] estimates suggested that 26 participants would be required in total (13 in each arm). Allowing for a potential dropout rate of 20%, 32 was the recruitment target for the study. Data are presented as mean ± SD or median (Q1–Q3) as appropriate. Generalised linear models (GLM) that adjust for baseline variability in each specified outcome were employed for comparison of change pre- and post-treatment visit measurements between the two groups for both primary and secondary endpoints. Where any of the GLM's statistical assumptions were not met, a non-parametric equivalent of GLM (using a b-spline, spline, or polynomial basis) was employed accordingly. The model response (outcome) was defined as the absolute change between visits and the model estimate (effect size) was adjusted for baseline (pre-treatment) values. Within-subject changes were determined using paired t-tests or Wilcoxon signed-rank test according to the distribution of data, which was assessed by Shapiro-Wilk test. These analyses were carried out using the per-protocol (PP) population tested at a significance level of 0.05. No adjustments were made for multiple testing.

Data were analysed using SAS 9.4 for Windows (Copyright © 2013, SAS Institute Inc., Cary, NC, USA) and STATA for Windows, StataCorp. 2021 (Stata Statistical Software: Release 17. College Station, TX: Stata-Corp LLC) and the PROC GLM command in SAS 9.4 for Windows

(Copyright© 2013, SAS Institute Inc., Cary, NC, USA). All recruited participants were included in the safety analysis and all participants who completed the study per protocol were included in the primary analysis. A data monitoring committee oversaw the study.

### Study approval
The clinical protocol received full ethical approval from the East of England Cambridge East Research Ethics Committee (reference 16/EE/0550). The clinical study was registered at clinicaltrials.gov, NCT03111810.

### Reporting summary
Further information on research design is available in the Nature Portfolio Reporting Summary linked to this article.

## Data availability
All data supporting the findings described in this manuscript are available in the article and in the Supplementary Information and from the corresponding author upon request. Source data are provided in this paper.

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

## Acknowledgements

The work was supported by the Medical Research Council (industry asset sharing initiative grant to J.W.T. ref. MR/N024591/1); NIHR Oxford Biomedical Research Centre (principal investigator award to J.W.T.); British Heart Foundation (senior fellowship to L.H. ref. FS/15/56/31645); University of Oxford/Novo Nordisk (Clinical research training fellowship awarded to A.M.). Exchange in Endocrinology Expertise Programme of the European Union of Medical Specialists (3E Fellowship awarded to R.P.), European Society of Endocrinology (short-term fellowship awarded to R.P.). R.H.S. is supported by the CSO. R.R.H. is an Emeritus NIHR Senior Investigator. The views expressed are those of the author(s) and not necessarily those of the NHS, the NIHR or the Department of Health.

## Author contributions

N.O.: formal analysis, investigation, writing – original draft. R.P.: formal analysis, investigation. A.A.: investigation. SW: investigation. I.B.: investigation. N.N.: investigation. A.M.: investigation. T.M.: investigation. R.H.S.: investigation. A.P.v.B.: investigation. M.v.F.: investigation. A.M.I.: writing – review and editing. E.B.: investigation. R.S.: investigation. F.K.: writing – review and editing. P.M.S.: writing – review and editing. C.W.: investigation. J.D.: investigation. R.E.: investigation. F.G.: investigation. T.C.: investigation. L.H.: writing – review and editing. K.J.E.: writing – review and editing. A.W.: writing – review and editing. U.K.: formal analysis. R.L.C.: formal analysis. C.A.B.S.: project administration. J.E.M.: project administration. O.A.: formal analysis. R.R.H.: writing – review and editing and J.W.T.: conceptualisation, methodology, resources, writing – review and editing, supervision, funding acquisition.

## Competing interests

J.W.T. has been an advisory board member for Lumos, Pfizer AstraZeneca and Poxel. R.R.H. reports research support from AstraZeneca, Bayer and Merck Sharp & Dohme, and personal fees from Anji Pharmaceuticals, Novartis and Novo Nordisk. A.M.I. received consultation fees, unconditional grants and hospitality to conferences from IBSA, Takeda and IPSEN. R.E. receives consultancy funding from Immunodiagnostic Systems, Sandoz, Samsung, Haoma Medica, CL Bio, Biocon, Coherus, Takeda, meeting presentations for Pharmacosmos, Alexion and Amgen, and grant funding from Roche, Pharmacosmos and Alexion. A.W., K.J.E. and U.K. are full-time employees of AstraZeneca. The work has been supported by the NIHR Oxford Biomedical Research Centre. The remaining authors declare no other competing interests.

## Additional information

Nantia Othonos [1], Riccardo Pofi [1,2], Anastasia Arvaniti[1,3], Sarah White[1], Ilaria Bonaventura[1,2], Nikolaos Nikolaou [1], Ahmad Moolla[1], Thomas Marjot[1,4], Roland H. Stimson [5], André P. van Beek [6], Martijn van Faassen[7], Andrea M. Isidori[2], Elizabeth Bateman[8], Ross Sadler[8], Fredrik Karpe [1], Paul M. Stewart[9], Craig Webster[10], Joanne Duffy[10], Richard Eastell[11], Fatma Gossiel[11], Thomas Cornfield[1], Leanne Hodson[1], K. Jane Escott[12], Andrew Whittaker[13], Ufuk Kirik[14], Ruth L. Coleman [1,15], Charles A. B. Scott[1,15], Joanne E. Milton[1,15], Olorunsola Agbaje[1,15], Rury R. Holman [1,15] & Jeremy W. Tomlinson [1] ✉

[1]Oxford Centre for Diabetes, Endocrinology and Metabolism, NIHR Oxford Biomedical Research Centre, University of Oxford, Churchill Hospital, Oxford OX3 7LE, UK. [2]Department of Experimental Medicine, Sapienza University of Rome, Viale Regina Elena, 324, 00161 Rome, Italy. [3]Department of Biological and Medical Sciences, Oxford Brookes University, Oxford OX3 0BP, UK. [4]Translational Gastroenterology Unit, NIHR Oxford Biomedical Research Centre, University of Oxford, John Radcliffe Hospital, Oxford OX3 9DU, UK. [5]University/BHF Centre for Cardiovascular Science, University of Edinburgh, Edinburgh EH16 4TJ, UK. [6]Department of Endocrinology, University of Groningen, University Medical Center Groningen, Groningen, the Netherlands. [7]Department of Laboratory Medicine, University of Groningen, University Medical Center Groningen, Groningen, the Netherlands. [8]Department of Immunology, Churchill Hospital, Oxford OX3 7LE, UK. [9]Faculty of Medicine & Health, University of Leeds, Clarendon Way, Leeds LS2 9NL, UK. [10]Department of Pathology, University Hospitals Birmingham, NHS Foundation Trust, Birmingham B15 2GW, UK. [11]Mellanby Centre for Musculoskeletal Research, Department of Oncology & Metabolism, Faculty of Medicine, Dentistry & Health, University of Sheffield, Sheffield SR10 2RX, UK. [12]Business Development & Licensing, BioPharmaceuticals R&D, AstraZeneca, Cambridge, UK. [13]Emerging Innovations Unit, Discovery Sciences, BioPharmaceuticals R&D, AstraZeneca, Cambridge, UK. [14]Quantitative Biology, Discovery Sciences, BioPharmaceuticals R&D AstraZeneca, Mölndal, Sweden. [15]Diabetes Trials Unit, Oxford Centre for Diabetes, Endocrinology and Metabolism, Churchill Hospital, Oxford OX3 7LJ, UK. ✉e-mail: Jeremy.tomlinson@ocdem.ox.ac.uk

