## [Peer Review File · Nature Communications]

11 β -HSD1 inhibition in men mitigates prednisolone-induced adverse effects in a proof-of-concept randomized double-blind placebo-controlled trialEditorial Note: This manuscript has been previously reviewed at another journal that is not operating a transparent peer review scheme. This document only contains reviewer comments and rebuttal letters for versions considered at *Nature Communications*.

REVIEWER COMMENT

Reviewer #2 (Remarks to the Author):

The revised manuscript is still fraught with errors and confusing statements.

Male only inclusion weak. Appropriate precautions could easily have been taken to include women of child-bearing potential and on OCs. Also if the authors state that the drug may not be effective in women/could not be used in women, then half the population would be excluded from any perceived benefit –no generalizability. Ethically inappropriate.

The authors make a very wild claim that reproductive risks are unknown for this drug so they excluded women. If this is indeed true the manufacturer should not be carrying out human trials without proper testing in animal models. Since others have studied this particular compound both men and women with contraceptive measures this statement is inaccurate and furthermore not ethical to state and if true then the drug could have adverse consequences for the age group of men who father children.

Use of Rd during low dose insulin infusion rationale is weak and not relevant. If this drug can only benefit those on low and not on high dose exogenous steroids, then what clinical use is it? The authors state that based on their prior data prednisolone does not inhibit RD at high dose insulin! – that is counter to all prior studies that have shown deleterious effects of steroids on Rd at high dose insulin – 1 μ /kg/min. at low dose insulin EGP effects predominate and not Rd.

The authors power this study on primary outcome...” we powered the study to detect a 20% difference in glucose disposal between the 2 Othonos et al. Response to reviewers arms of the study from a typical mean (\pm 1SD) baseline of 2.91 ± 0.52 mg/kg/min (PMID:32594135) with a power of 0.8 and a type 1 error of 0.05; estimates suggested that 26 participants would be required in total (13 in each arm).....but yet mention in their response to reviewer 2 that “....it is possible that the study was underpowered to detect changes in glucose disposal”.

The above two statements are completely contradictory and frankly are confusing.

The responses to prior critique are not satisfactory. This same drug has been shown in prior studies to not change hepatic insulin sensitivity in humans- how do the authors explain their results?

Reviewer #3 (Remarks to the Author):

I do believe that most points raised by the Reviewers have been answered adequately.

Reviewer #4 (Remarks to the Author):

This is an adequately performed clinical trial and a well-written paper. The statistical analysis is done correctly and the conclusions are justified.

Reviewer #5 (Remarks to the Author):

General view:

The authors reported the effects of the 11-B-HSD-1 inhibitor on mitigating the adverse effects of prednisolone therapy on glucose disposal (Gd), circulating lipid profiles, osteocalcin, urinary steroid metabolites, and gene expression. This is a proof of concept very early pilot study in a small sample size of healthy men on the clinically significant concept, and the manuscript is well structured. There are a few minor comments on the statistical approaches and reporting.

Title:

It is suggested that the manuscript should have Identification as a pilot/proof of concept randomised trial in the title and abstract.

Abstract:

The number and type of adverse events should be mentioned in the abstract.

Methods:

- GLM usually accounts for Generalised linear models. Based on the type of the outcome, the authors should clearly explain which model was used for data analysis and which Macro or Commands.
- Which baseline variables were included in the final model for analysis?
- The type of tests and final models used for analysis should be evident in the tables.

Results:

- It was mentioned that median (IQR) was used in the Tables; however, it sounds like results were reported as median (Q1 to Q3).
- It is unclear whether the reported effects in Table2 are crude or adjusted.

Discussion:

Age range and its effect on the generalisability of the results and observed adverse events should be discussed.

Reviewer #1

No additional comments provided

Reviewer #2

The revised manuscript is still fraught with errors and confusing statements. Male only inclusion weak. Appropriate precautions could easily have been taken to include women of child-bearing potential and on OCs. Also if the authors state that the drug may not be effective in women/could not be used in women, then half the population would be excluded from any perceived benefit –no generalizability. Ethically inappropriate.

The authors make a very wild claim that reproductive risks are unknown for this drug so they excluded women. If this is indeed true the manufacturer should not be carrying out human trials without proper testing in animal models. Since others have studied this particular compound both men and women with contraceptive measures this statement is inaccurate and furthermore not ethical to state and if true then the drug could have adverse consequences for the age group of men who father children.

We do acknowledge that now there are data that have been published where AZD4017 has been given to women of childbearing age where adequate contraception has been used). This paper has been cited in the revised manuscript (reference number 9). We also agree that the phrasing of the sentence about reproductive toxicity could be misleading, and we have amended this in the methods (participants) section.

As we state in the manuscript, this is a proof-of-concept study and we had wanted to keep the participant population under investigations as homogeneous as possible and this was an additional factor that we took into consideration when deciding to recruit male participants only. Standardizing to time of investigations around the timing of the menstrual cycle or adjusting for the use of hormonal contraceptive pills which can impact upon metabolic and bone readouts would also have provided significant additional challenges. As mentioned above, we have repeatedly emphasised (title, abstract, methods, results and discussion) that this is a study in male participants only and that this can limit interpretation and there would be a need for confirmatory studies in female participants.

Use of Rd during low dose insulin infusion rationale is weak and not relevant. If this drug can only benefit those on low and not on high dose exogenous steroids, then what clinical use is it? The authors state that based on their prior data prednisolone does not inhibit RD at high dose insulin! – that is counter to all prior studies that have shown deleterious effects of steroids on Rd at high dose insulin – 1 mu/kg/min. at low dose insulin EGP effects predominate and not Rd.

We apologize if there was some confusion in our original response to the reviewer comments. Our previous study had used a lower dose of prednisolone (10mg, contrasting with 20mg in this study). In the current study, we do show that prednisolone decreases Rd at high dose insulin in agreement with the published literature (placebo arm, table 2).

An important consideration is that we have used relatively high doses of insulin as part of our 2-step clamps (20mU/m²/min [\sim 0.5mu/kg/min] and 100mU/m²/min [\sim 2.0mu/kg/min]) based upon the premise that we would be making the participants insulin resistant with prednisolone treatment. As such, the 'low dose' insulin is not very 'low' and therefore there will be both effect on EGP and Rd and this justified our approach to use this as the primary end-point.

The very high doses of insulin that were used, might have meant that the observed magnitude of the effect of prednisolone, whilst still present (table 2), may have been blunted as the impact of the high dose insulin effect was so large. This might therefore have compromised the ability to detect the benefit of 11 β -HSD1 inhibition and therefore, as part of the discussion, we suggest that future studies may use alternative assessments of glucose handling and glucose homeostasis.

The authors power this study on primary outcome..." we powered the study to detect a 20% difference

in glucose disposal between the 2 Othonos et al. Response to reviewers arms of the study from a typical mean (± 1 SD) baseline of 2.91 ± 0.52 mg/kg/min (PMID:32594135) with a power of 0.8 and a type 1 error of 0.05; estimates suggested that 26 participants would be required in total (13 in each arm)”. . . .but yet mention in their response to reviewer 2 that “. . .it is possible that the study was underpowered to detect changes in glucose disposal”.

The above two statements are completely contradictory and frankly are confusing.

Again, we apologise for the confusion. We did use the cited study to base our power and samples size calculations, but the variability of baseline glucose disposal (Gd) in the current study was higher than we had observed in our previous study. It was this unexpected, and higher than anticipated, variability in Gd that meant that the study was underpowered. We have now clarified this in the discussion.

The responses to prior critique are not satisfactory. This same drug has been shown in prior studies to not change hepatic insulin sensitivity in humans- how do the authors explain their results?

The reviewer is correct that AZD4017 has been shown not to impact on hepatic insulin sensitivity, but that study is in an entirely different clinical context (NAFLD) (Yadav et al. 2022, <https://doi.org/10.1111/dom.14646>, already cited within the current manuscript). Our study addresses the impact of prescribed glucocorticoids and mitigating their effects on metabolic phenotype as opposed to treating metabolic disease *per se*. Therefore, comparisons cannot be drawn with the current paper as it addresses a completely different clinical question.

Reviewer #3

I do believe that most points raised by the Reviewers have been answered adequately.

We are pleased that the reviewer is happy with the responses that were provided.

Reviewer #4:

This is an adequately performed clinical trial and a well-written paper. The statistical analysis is done correctly and the conclusions are justified.

We thank the reviewer for their positive comments.

Reviewer #5:

General view:

The authors reported the effects of the 11-B-HSD-1 inhibitor on mitigating the adverse effects of prednisolone therapy on glucose disposal (Gd), circulating lipid profiles, osteocalcin, urinary steroid metabolites, and gene expression. This is a proof of concept very early pilot study in a small sample size of healthy men on the clinically significant concept, and the manuscript is well structured. There are a few minor comments on the statistical approaches and reporting.

Title:

It is suggested that the manuscript should have Identification as a pilot/proof of concept randomised trial in the title and abstract.

We have added that this was a proof-of-concept in the title, abstract and the methods.

Abstract:

The number and type of adverse events should be mentioned in the abstract.

These have now been included in the abstract.

Methods:

•GLM usually accounts for Generalised linear models. Based on the type of the outcome, the authors should clearly explain which model was used for data analysis and which Macro or Commands.

- Which baseline variables were included in the final model for analysis?
- The type of tests and final models used for analysis should be evident in the tables.

We have clarified the statistical analysis section within the methods to confirm that Generalised Linear Models (GLM) were used for all primary and secondary endpoints. This was undertaken using the SAS statistical package and the PROC GLM commands. This has now also now been included in the statistical analysis section. The GLM takes into account the variability in baseline measurements for each of the specified outcomes (primary, secondary or exploratory). Again, we have clarified this in the statistical methods section. We have now also added to the table legend to include the details of which statistical analyses were undertaken.

Results:

- It was mentioned that median (IQR) was used in the Tables; however, it sounds like results were reported as median (Q1 to Q3).

The reviewer is correct, and we apologise for this. Data are reported as Q1 to Q3 and we have now amended the table legends and methods section.

- It is unclear whether the reported effects in Table2 are crude or adjusted.

The effects reported in Table 2 are all adjusted for baseline (pre-treatment) values for each specified outcome) and we have now added clarification to the table legend.

Discussion:

Age range and its effect on the generalisability of the results and observed adverse events should be discussed.

We have now added to the discussion to state that the relatively young age of the cohort (mean age = 38 years) is a limitation, and that studies in more elderly populations, including those taking prescribed glucocorticoids for inflammatory conditions are now needed.

REVIEWERS' COMMENTS

Reviewer #5 (Remarks to the Author):

Thank you for considering the comments and providing point-by-point responses.

I do believe that most of the points raised before have been appropriately answered. There are only a few minor comments on the revised version :

- The authors confirmed that they had used Generalised linear models, but in line#540, GLM still stands for Generalised non-linear models (GLM).
- Because the PROC GLM is only applicable in SAS, It might be better to change lines#551 to #553 as below:

Data were analysed using Stata for Windows, StataCorp. 2021 (Stata Statistical Software: Release 17. College StationTX: StataCorp LLC) and the PROC GLM command in SAS 9.4 for Windows (Copyright © 2013, SAS Institute Inc., Cary, NC, USA).

Reviewer #5:

•The authors confirmed that they had used Generalised linear models, but in line#540, GLM still stands for Generalised non-linear models (GLM).

We apologise, and this has now been corrected.

•Because the PROC GLM is only applicable in SAS, It might be better to change lines#551 to #553 as below:

Data were analysed using Stata for Windows, StataCorp. 2021 (Stata Statistical Software: Release 17. College StationTX: StataCorp LLC) and the PROC GLM command in SAS 9.4 for Windows (Copyright ©2013, SAS Institute Inc., Cary, NC, USA)

We have amended the manuscript as suggested by the reviewer.